# DynaBERT: Dynamic BERT with Adaptive Width and Depth

**Lu Hou[1], Zhiqi Huang[2], Lifeng Shang[1], Xin Jiang[1], Xiao Chen[1], Qun Liu[1]**
[1]Huawei Noah's Ark Lab
{houlu3,shang.lifeng,Jiang.Xin,chen.xiao2,qun.liu}@huawei.com
[2]Peking University, China
zhiqihuang@pku.edu.cn

## Abstract

The pre-trained language models like BERT, though powerful in many natural language processing tasks, are both computation and memory expensive. To alleviate this problem, one approach is to compress them for specific tasks before deployment. However, recent works on BERT compression usually compress the large BERT model to a fixed smaller size. They can not fully satisfy the requirements of different edge devices with various hardware performances. In this paper, we propose a novel dynamic BERT model (abbreviated as DynaBERT), which can flexibly adjust the size and latency by selecting adaptive width and depth. The training process of DynaBERT includes first training a width-adaptive BERT and then allowing both adaptive width and depth, by distilling knowledge from the full-sized model to small sub-networks. Network rewiring is also used to keep the more important attention heads and neurons shared by more sub-networks. Comprehensive experiments under various efficiency constraints demonstrate that our proposed dynamic BERT (or RoBERTa) at its largest size has comparable performance as BERT$_{\text{BASE}}$ (or RoBERTa$_{\text{BASE}}$), while at smaller widths and depths consistently outperforms existing BERT compression methods. Code is available at https://github.com/huawei-noah/Pretrained-Language-Model/tree/master/DynaBERT.

## 1 Introduction

Recently, pre-trained language models based on the Transformer [24] structure like BERT [5] and RoBERTa [14] have achieved remarkable results on natural language processing tasks. However, these models have many parameters, hindering their deployment on edge devices with limited storage, computation, and energy consumption. The difficulty of deploying BERT to these devices lies in two aspects. Firstly, the hardware performances of various devices vary a lot, and it is infeasible to deploy one single BERT model to all kinds of edge devices. Thus different architectural configurations of the BERT model are desired. Secondly, the resource condition of one device under different circumstances can be quite different. For instance, on a mobile phone, when a large number of compute-intensive or storage-intensive programs are running, the resources that can be allocated to the current BERT model will be correspondingly fewer. Thus once the BERT model is deployed, dynamically selecting a part of the model (also referred to as *sub-networks*) for inference based on the device's current resource condition is also desirable. Note that unless otherwise specified, the BERT model mentioned in this paper refers to a task-specific BERT rather than the pre-trained model.

There have been some attempts to compress and accelerate inference of the Transformer-based models using low-rank approximation [15, 12], weight-sharing [4, 12], knowledge distillation [21, 23, 10, 28], quantization [1, 33, 22, 8] and pruning [16, 17, 3, 25]. However, these methods usually compress the

model to a fixed size and can not meet the requirements above. In [4, 7, 6, 13, 29, 34], Transformer-based models with adaptive depth are proposed to dynamically select some of the Transformer layers during inference. However, these models only consider compression in the depth direction and generate a limited number of architectural configurations, which can be restrictive for various deployment requirements. Some studies now show that the width direction also has high redundancy. For example, in [25, 17], it is shown that only a small number of attention heads are required to keep comparable accuracy. There have been some works that train convolutional neural networks (CNNs) with adaptive width [32, 31, 30], and even both adaptive width and depth [2]. However, since each Transformer layer in the BERT model includes both a Multi-Head Attention (MHA) module and a position-wise Feed-forward Network (FFN) that perform transformations in two different dimensions (i.e., the sequence and the feature dimensions), the width of the BERT model can not be simply defined as the number of kernels as in CNNs. Moreover, successive training of first along depth and then width as in [2] can be sub-optimal since these two directions are hard to disentangle. This may also cause the knowledge learned in the depth direction to be forgotten after the width is trained to be adaptive.

In this work, we propose a novel DynaBERT model that offers flexibility in both width and depth directions of the BERT model. Compared to [4, 7, 6, 13] where only depth is adaptive, DynaBERT enables a significantly richer number of architectural configurations and better exploration of the balance between model accuracy and size. Concurrently to our work, flexibility in both directions is also proposed in [27], but on the encoder-decoder Transformer structure [24] and on machine translation task. Besides the difference in the model and task, our proposed DynaBERT also advances in the following aspects: (1) We distill knowledge from the full-sized teacher model to smaller student sub-networks to reduce the accuracy drop caused by the lower capacity of smaller size. (2) Before allowing both adaptive width and depth, we train an only width-adaptive BERT (abbreviated as DynaBERT$_W$) to act as a teacher assistant to bridge the large gap of model size between the student and teacher. (3) For DynaBERT$_W$, we rewire the connections in each Transformer layer to ensure that the more important heads and neurons are utilized by more sub-networks. (4) Once DynaBERT is trained, no further fine-tuning is required for each sub-network. Extensive experiments on the GLUE benchmark and SQuAD under various efficiency constraints show that, our proposed dynamic BERT (or RoBERTa) at its largest size performs comparably as BERT$_{BASE}$ (or RoBERTa$_{BASE}$), while at smaller sizes outperforms other BERT compression methods.

## 2 Method

In this section, we elaborate on the training method of our DynaBERT model. The training process (Figure 1) includes two stages. We first train a width-adaptive DynaBERT$_W$ in Section 2.1 and then train the both width- and depth-adaptive DynaBERT in Section 2.2. Directly using the knowledge distillation to train DynaBERT without DynaBERT$_W$, or first train a depth-adaptive BERT and then distill knowledge from it to DynaBERT leads to inferior performance (Details are in Section 3.3).

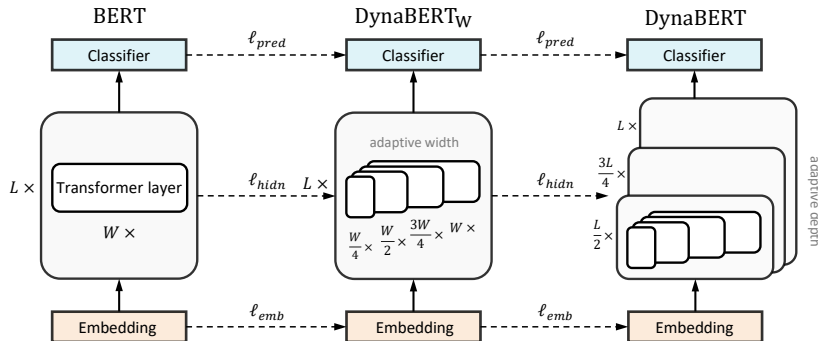

Figure 1: A two-stage procedure to train DynaBERT. First, using knowledge distillation (dashed lines) to transfer the knowledge from a fixed teacher model to student sub-networks with adaptive width in DynaBERT$_W$. Then, using knowledge distillation (dashed lines) to transfer the knowledge from a trained DynaBERT$_W$ to student sub-networks with adaptive width and depth in DynaBERT.

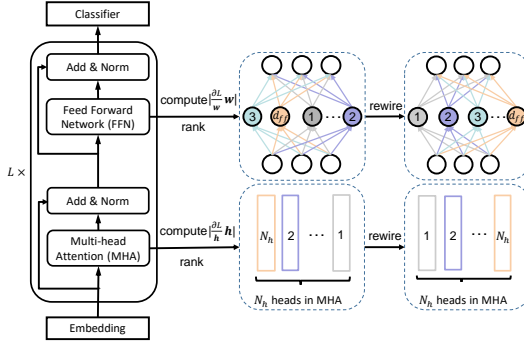

Figure 2: Rewire connections in BERT based on the importance of attention heads in MHA and neurons in the intermediate layer of FFN.

**Algorithm 1** Train DynaBERT$_W$ or DynaBERT.

1: **if** training DynaBERT$_W$ **then**
2:     $\mathcal{L}\leftarrow(3)$, $InitM\leftarrow$rewired net, $depthList$=[1].
3: **else**:
4:     $\mathcal{L}\leftarrow(4)$, $InitM\leftarrow$DynaBERT$_W$.
5: **initialize** a fixed teacher model and a trainable student model with $InitM$.
6: **for** $iter = 1,\cdots,T_{train}$ **do**
7:     Get next mini-batch of training data.
8:     Clear gradients in the student model.
9:     **for** $m_d$ in $depthList$ **do**
10:         **for** $m_w$ in $widthList$ **do**
11:             Compute loss $\mathcal{L}$.
12:             Accumulate gradient $\mathcal{L}.backward()$.
13:         **end for**
14:     **end for**
15:     Update with the accumulated gradients.
16: **end for**

## 2.1 Training DynaBERT$_W$ with Adaptive Width

Compared to CNNs stacked with regular convolutional layers, the BERT model is built with Transformer layers, and the width of it can not be trivially determined due to the more complicated computation involved. Specifically, a standard Transformer layer contains a Multi-Head Attention (MHA) layer and a Feed-Forward Network (FFN). In the following, we rewrite the original formulation of MHA and FFN in [24] in a different way to show that, the computation of the attention heads of MHA and the neurons in the intermediate layers of FFN can be performed in parallel. Thus we can adjust the width of a Transformer layer by varying the number of attention heads and neurons in the intermediate layer of FFN.

For the $t$-th Transformer layer, suppose the input to it is $\mathbf{X} \in \mathbb{R}^{n\times d}$ where $n$ and $d$ are the sequence length and hidden state size, respectively. Following [17], we divide the computation of the MHA into the computations for each attention head. suppose there are $N_H$ attention heads in each layer, with head $h$ parameterized by $\mathbf{W}_h^Q, \mathbf{W}_h^K, \mathbf{W}_h^V, \mathbf{W}_h^O \in \mathbb{R}^{d\times d_h}$ where $d_h = d/N_H$. The output of the head $h$ is computed as $\text{Attn}_{\mathbf{W}_h^Q,\mathbf{W}_h^K,\mathbf{W}_h^V,\mathbf{W}_h^O}^h(\mathbf{X}) = \text{Softmax}(\frac{1}{\sqrt{d}}\mathbf{X}\mathbf{W}_h^Q\mathbf{W}_h^{K\top}\mathbf{X}^\top)\mathbf{X}\mathbf{W}_h^V\mathbf{W}_h^{O\top}$. In multi-head attention, $N_H$ heads are computed in parallel to get the final output [25]:

$$\text{MHAttn}_{\mathbf{W}^Q,\mathbf{W}^K,\mathbf{W}^V,\mathbf{W}^O}(\mathbf{X}) = \sum_{h=1}^{N_H} \text{Attn}_{\mathbf{W}_h^Q,\mathbf{W}_h^K,\mathbf{W}_h^V,\mathbf{W}_h^O}^h(\mathbf{X}). \tag{1}$$

Suppose the two linear layers in FFN are parameterized by $\mathbf{W}^1 \in \mathbb{R}^{d\times d_{ff}}, \mathbf{b}^1 \in \mathbb{R}^{d_{ff}}$ and $\mathbf{W}^2 \in \mathbb{R}^{d_{ff}\times d}, \mathbf{b}^2 \in \mathbb{R}^d$, where $d_{ff}$ is the number of neurons in the intermediate layer of FFN. Denote the input of FFN is $\mathbf{A} \in \mathbb{R}^{n\times d}$, the output of FFN can be divided into computations of $d_{ff}$ neurons:

$$\text{FFN}_{\mathbf{W}^1,\mathbf{W}^2,\mathbf{b}^1,\mathbf{b}^2}(\mathbf{A}) = \sum_{i=1}^{d_{ff}} \text{GeLU}(\mathbf{A}\mathbf{W}_{:,i}^1 + b_i^1)\mathbf{W}_{i,:}^2 + \mathbf{b}^2. \tag{2}$$

Based on (1) and (2), the width of a Transformer layer can be adapted by varying the number of attention heads in MHA and neurons in the intermediate layer of FFN (Figure 2). For width multiplier $m_w$, we retain the leftmost $\lfloor m_w N_H \rfloor$ attention heads in MHA and $\lfloor m_w d_{ff} \rfloor$ neurons in the intermediate layer of FFN. In this case, each Transformer layer is roughly compressed by a ratio of $m_w$. This is not strictly equal as layer normalization and bias in linear layers also have very few parameters. The number of neurons in the embedding dimension is not adapted because these neurons are connected through skip connections across all Transformer layers and cannot be flexibly scaled.

### 2.1.1 Network Rewiring

To fully utilize the network's capacity, the more important heads or neurons should be shared across more sub-networks. Thus before training the width-adaptive network, we rank the attention heads and neurons according to their importance in the fine-tuned BERT model, and then arrange them with descending importance in the width direction (Figure 2).

Following [18, 25], we compute the importance score of a head or neuron based on the variation in the training loss $\mathcal{L}$ if we remove it. Specifically, for one head with output $\mathbf{h}$, its importance $I_h$ can be estimated using the first-order Taylor expansion as

$$I_{\mathbf{h}} = |\mathcal{L}_{\mathbf{h}} - \mathcal{L}_{\mathbf{h}=\mathbf{0}}| = \left| \mathcal{L}_{\mathbf{h}} - \left( \mathcal{L}_{\mathbf{h}} - \frac{\partial \mathcal{L}}{\partial \mathbf{h}}(\mathbf{h} - \mathbf{0}) + R_{\mathbf{h}=\mathbf{0}} \right) \right| \approx \left| \frac{\partial \mathcal{L}}{\partial \mathbf{h}} \mathbf{h} \right|$$

if we ignore the remainder $R_{\mathbf{h}=\mathbf{0}}$. Similarly, for a neuron in the intermediate layer of FFN, denote the set of weights in $\mathbf{W}^1$ and $\mathbf{W}^2$ connected to it as $\mathbf{w} = \{w_1, w_2, \cdots, w_{2d}\}$, its importance is estimated by

$$\left| \frac{\partial \mathcal{L}}{\partial \mathbf{w}} \mathbf{w} \right| = \left| \sum\nolimits_{i=1}^{2d} \frac{\partial \mathcal{L}}{\partial w_i} w_i \right|.$$

Empirically, we use the development set to calculate the importance of attention heads and neurons.

### 2.1.2 Training with Adaptive Width

After the connections of the BERT model are rewired according to Section 2.1.1, we use knowledge distillation to train DynaBERT$_\mathrm{W}$. Specifically, we use the rewired BERT model as the fixed teacher network, and to initialize DynaBERT$_\mathrm{W}$. Then we distill the knowledge from the fixed teacher model to student sub-networks at different widths in DynaBERT$_\mathrm{W}$ (First stage in Figure 1).

Following [10], we transfer the knowledge in logits $\mathbf{y}$, embedding (i.e., the output of the embedding layer) $\mathbf{E}$, and hidden states (i.e. the output of each Transformer layer) $\mathbf{H}_l$ ($l = 1, 2, \cdots, L$) from the teacher model to $\mathbf{y}^{(m_w)}, \mathbf{E}^{(m_w)}$ and $\mathbf{H}_l^{(m_w)}$ of the student sub-network with width multiplier $m_w$. Here $\mathbf{E}, \mathbf{H}_l, \mathbf{E}^{(m_w)}, \mathbf{H}_l^{(m_w)} \in \mathbb{R}^{n \times d}$. Denote SCE as the soft cross-entropy loss and MSE as the mean squared error. The three distillation loss terms are

$$\ell_{pred}(\mathbf{y}^{(m_w)}, \mathbf{y}) = \mathrm{SCE}(\mathbf{y}^{(m_w)}, \mathbf{y}), \quad \ell_{emb}(\mathbf{E}^{(m_w)}, \mathbf{E}) = \mathrm{MSE}(\mathbf{E}^{(m_w)}, \mathbf{E}),$$

$$\ell_{hidn}(\mathbf{H}^{(m_w)}, \mathbf{H}) = \sum\nolimits_{l=1}^{L} \mathrm{MSE}(\mathbf{H}_l^{(m_w)}, \mathbf{H}_l).$$

Thus the training objective is

$$\mathcal{L} = \lambda_1 \ell_{pred}(\mathbf{y}^{(m_w)}, \mathbf{y}) + \lambda_2 (\ell_{emb}(\mathbf{E}^{(m_w)}, \mathbf{E}) + \ell_{hidn}(\mathbf{H}^{(m_w)}, \mathbf{H})), \tag{3}$$

where $\lambda_1$ and $\lambda_2$ are the scaling parameters that control the weights of different loss terms. Note that we use the same scaling parameter for the distillation loss of the embedding and hidden states, because the two have the same dimension and similar scale. Empirically, we choose $(\lambda_1, \lambda_2) = (1, 0.1)$ because $\ell_{emb} + \ell_{hidn}$ is about one magnitude larger than $\ell_{pred}$. The detailed training process of DynaBERT$_\mathrm{W}$ is shown in Algorithm 1, where we restrict the depth multiplier $m_d$ to 1 (i.e., the largest depth) as DynaBERT$_\mathrm{W}$ is only adaptive in width. To provide more task-specific data for distillation learning, we use the data augmentation method in TinyBERT [10], which uses a pre-trained BERT [5] trained from the masked language modeling task to generate task-specific augmented samples.

## 2.2 Training DynaBERT with Adaptive Width and Depth

After DynaBERT$_\mathrm{W}$ is trained, we use it as the fixed teacher model, and to initialize the DynaBERT model. Then we distill the knowledge from the fixed teacher model at the maximum depth to student sub-networks at equal or lower depths (Second stage in Figure 1). To avoid catastrophic forgetting of learned elasticity in the width direction, we still train over different widths in each iteration. For width multiplier $m_w$, the objective of the student sub-network with depth multiplier $m_d$ still contains three terms $\ell'_{pred}, \ell'_{emb}$ and $\ell'_{hidn}$ as in (3). It makes the logits $\mathbf{y}^{(m_w, m_d)}$, embedding $\mathbf{E}^{(m_w, m_d)}$ and hidden states $\mathbf{H}^{(m_w, m_d)}$ mimic $\mathbf{y}^{(m_w)}, \mathbf{E}^{(m_w)}$ and $\mathbf{H}^{(m_w)}$ from the teacher model with the maximum depth. When the depth multiplier $m_d < 1$, the student has fewer layers than the teacher. In this case, we use the "Every Other" strategy in [7] and drop layers evenly to get a balanced network. Then we match the hidden states of the remaining layers $L_S$ in the student sub-network with those at depth $d \in L_T$ which satisfies $\mathrm{mod}(d + 1, \frac{1}{1 - m_d}) \neq 0$ from the teacher model as

$$\ell'_{hidn}(\mathbf{H}^{(m_w, m_d)}, \mathbf{H}^{(m_w)}) = \sum\nolimits_{l, l' \in L_S, L_T} \mathrm{MSE}(\mathbf{H}_l^{(m_w, m_d)}, \mathbf{H}_{l'}^{(m_w)}).$$

We use $d + 1$ here because we want to keep the knowledge in the last layer of the teacher model which is shown to be important in [28]. A detailed example can be found at Appendix A. Thus the distillation objective can still be written as

$$\mathcal{L} = \lambda_1 \ell'_{pred}(\mathbf{y}^{(m_w, m_d)}, \mathbf{y}^{(m_w)}) + \lambda_2 (\ell'_{emb}(\mathbf{E}^{(m_w, m_d)}, \mathbf{E}^{(m_w)}) + \ell'_{hidn}(\mathbf{H}^{(m_w, m_d)}, \mathbf{H}^{(m_w)})). \quad (4)$$

For simplicity, we do not tune $\lambda_1, \lambda_2$ and choose $(\lambda_1, \lambda_2) = (1, 1)$ in our experiments. The training procedure can be found in Algorithm 1. After training with the augmented data and the distillation objective above (Step 1), one can further fine-tune the network using the original data and the cross-entropy loss between the predicted labels and ground-truth labels (Step 2). Step 2 further improves the performance on some data sets empirically (Details are in Section 3.3). In this work, we report results of the model with higher average validation accuracy of all sub-networks, between before (Step 1) and after fine-tuning (Step 2) with the original data.

## 3 Experiment

In this section, we evaluate the efficacy of the proposed DynaBERT on the General Language Understanding Evaluation (GLUE) tasks [26] and the machine reading comprehension task SQuAD v1.1 [19], using both BERT$_{BASE}$ [5] and RoBERTa$_{BASE}$ [14] as the backbone models. The corresponding both width- and depth-adaptive BERT and RoBERTa models are named as DynaBERT and DynaRoBERTa, respectively. For BERT$_{BASE}$ and RoBERTa$_{BASE}$, the number of Transformer layers is $L = 12$, the hidden state size is $d = 768$. In each Transformer layer, the number of heads in MHA is $N_H = 12$, and the number of neurons in the intermediate layer in FFN is $d_{ff} = 3072$. The list of width multipliers is $[1.0, 0.75, 0.5, 0.25]$, and the list of depth multipliers is $[1.0, 0.75, 0.5]$. There are a total of $4 \times 3 = 12$ different configurations of sub-networks. We use Nvidia V100 GPU for training. Detailed hyperparameters for the experiments are in Appendix B.2.

We compare the proposed DynaBERT and DynaRoBERTa with (i) the base models BERT$_{BASE}$ [5] and RoBERTa$_{BASE}$ [14]; and (ii) popular BERT compression methods, including distillation methods DistilBERT [21], TinyBERT [10], and adaptive-depth method LayerDrop [7]. The results of the compared methods are taken from their original paper or official code repository. We evaluate the efficacy of our proposed DynaBERT and DynaRoBERTa under different efficiency constraints, including #parameters, FLOPs, the latency on Nvidia K40 GPU and Kirin 810 A76 ARM CPU (Details can be found in Appendix B.3).

### 3.1 Results on the GLUE benchmark

**Data.** The GLUE benchmark [26] is a collection of diverse natural language understanding tasks. Detailed descriptions of GLUE data sets are included in Appendix B.1. Following [5], for the development set, we report Spearman correlation for `STS-B`, Matthews correlation for `CoLA` and accuracy for the other tasks. For the test set of `QQP` and `MRPC`, we report "F1".

**Main Results.** Table 1 shows the results of sub-networks derived from the proposed DynaBERT and DynaRoBERTa. The proposed DynaBERT (or DynaRoBERTa) achieves comparable performances as BERT$_{BASE}$ (or RoBERTa$_{BASE}$) with the same or smaller size. For most tasks, the sub-network of DynaBERT or DynaRoBERTa with the maximum size does not necessarily have the best performance, indicating that redundancy exists in the original BERT or RoBERTa model. Indeed, with the proposed method, the model's width and depth for most tasks can be reduced without performance drop. Another observation is that using one specific width multiplier usually has higher accuracy than using the same depth multiplier. This indicates that compared to the depth direction, the width direction is more robust to compression. Sub-networks from DynaRoBERTa, most of the time, perform significantly better than those from DynaBERT under the same depth and width. Test set results in Appendix C.1 also show that DynaBERT (resp. DynaRoBERTa) at its largest size has comparable or better accuracy as BERT$_{BASE}$ (resp. RoBERTa$_{BASE}$).

**Comparison with Other Methods.** Figure 3 compares DynaBERT and DynaRoBERTa on `SST-2` and `MNLI` with other methods under different efficiency constraints, i.e., #parameters, FLOPs, latency on Nvidia K40 GPU and Kirin 810 ARM CPU. Results of the other data sets are in Appendix C.1. Note that each number of TinyBERT and DistilBERT uses a different model, while different numbers of our proposed DynaBERT and DynaRoBERTa use different sub-networks within one model.

Table 1: Development set results of the GLUE benchmark using DynaBERT and DynaRoBERTa with different width and depth multipliers $(m_w, m_d)$.

| Method | CoLA | | | STS-B | | | MRPC | | | RTE | | |
|---|---|---|---|---|---|---|---|---|---|---|---|---|
| BERT$_{BASE}$ | 58.1 | | | 89.8 | | | 87.7 | | | 71.1 | | |
| $m_w$\\$m_d$ | 1.0x | 0.75x | 0.5x | 1.0x | 0.75x | 0.5x | 1.0x | 0.75x | 0.5x | 1.0x | 0.75x | 0.5x |
| DynaBERT 1.0x | 59.7 | 59.1 | 54.6 | **90.1** | 89.5 | 88.6 | 86.3 | 85.8 | 85.0 | 72.2 | 71.8 | 66.1 |
| 0.75x | **60.8** | 59.6 | 53.2 | 90.0 | 89.4 | 88.5 | **86.5** | 85.5 | 84.1 | 71.8 | **73.3** | 65.7 |
| 0.5x | 58.4 | 56.8 | 48.5 | 89.8 | 89.2 | 88.2 | 84.8 | 84.1 | 83.1 | 72.2 | 72.2 | 67.9 |
| 0.25x | 50.9 | 51.6 | 43.7 | 89.2 | 88.3 | 87.0 | 83.8 | 83.8 | 81.4 | 68.6 | 68.6 | 63.2 |

| Method | MNLI- (m/mm) | | | QQP | | | QNLI | | | SST-2 | | |
|---|---|---|---|---|---|---|---|---|---|---|---|---|
| BERT$_{BASE}$ | 84.8/84.9 | | | 90.9 | | | 92.0 | | | 92.9 | | |
| $m_w$\\$m_d$ | 1.0x | 0.75x | 0.5x | 1.0x | 0.75x | 0.5x | 1.0x | 0.75x | 0.5x | 1.0x | 0.75x | 0.5x |
| DynaBERT 1.0x | **84.9/85.5** | 84.4/85.1 | 83.7/84.6 | **91.4** | **91.4** | 91.1 | 92.1 | 91.7 | 90.6 | 93.2 | 93.3 | 92.7 |
| 0.75x | 84.7/85.5 | 84.3/85.2 | 83.6/84.4 | **91.4** | 91.3 | 91.2 | 92.2 | 91.8 | 90.7 | 93.0 | 93.1 | 92.8 |
| 0.5x | 84.7/85.2 | 84.2/84.7 | 83.0/83.6 | 91.3 | 91.2 | 91.0 | **92.2** | 91.5 | 90.0 | **93.3** | 92.7 | 91.6 |
| 0.25x | 83.9/84.2 | 83.4/83.7 | 82.0/82.3 | 90.7 | 91.1 | 90.4 | 91.5 | 90.8 | 88.5 | 92.8 | 92.0 | 92.0 |

| Method | CoLA | | | STS-B | | | MRPC | | | RTE | | |
|---|---|---|---|---|---|---|---|---|---|---|---|---|
| RoBERTa$_{BASE}$ | 65.1 | | | 91.2 | | | 90.7 | | | 81.2 | | |
| $m_w$\\$m_d$ | 1.0x | 0.75x | 0.5x | 1.0x | 0.75x | 0.5x | 1.0x | 0.75x | 0.5x | 1.0x | 0.75x | 0.5x |
| DynaRoBERTa 1.0x | 63.6 | 61.0.7 | 59.5 | **91.3** | 91.0 | 90.0 | 88.7 | 89.7 | 88.5 | **82.3** | 78.7 | 72.9 |
| 0.75x | **63.7** | 61.4 | 54.9 | 91.0 | 90.7 | 89.7 | 90.0 | 89.2 | 88.2 | 79.4 | 77.3 | 70.8 |
| 0.5x | 61.3 | 58.1 | 52.9 | 90.3 | 90.1 | 88.9 | **90.4** | 90.0 | 86.5 | 75.1 | 73.6 | 71.5 |
| 0.25x | 54.2 | 46.7 | 39.8 | 89.6 | 89.2 | 87.5 | 88.2 | 88.0 | 84.3 | 70.0 | 70.0 | 66.8 |

| Method | MNLI- (m/mm) | | | QQP | | | QNLI | | | SST-2 | | |
|---|---|---|---|---|---|---|---|---|---|---|---|---|
| RoBERTa$_{BASE}$ | 87.5/87.5 | | | 91.8 | | | 93.1 | | | 95.2 | | |
| $m_w$\\$m_d$ | 1.0x | 0.75x | 0.5x | 1.0x | 0.75x | 0.5x | 1.0x | 0.75x | 0.5x | 1.0x | 0.75x | 0.5x |
| DynaRoBERTa 1.0x | **88.3/87.6** | 87.7/87.2 | 86.2/85.8 | 92.0 | **92.0** | 91.7 | **92.9** | 92.5 | 91.4 | **95.1** | 94.3 | 93.3 |
| 0.75x | 88.0/87.3 | 87.5/86.7 | 85.8/85.4 | 91.9 | 91.8 | 91.6 | 92.8 | 92.4 | 91.3 | 94.6 | 94.3 | 93.3 |
| 0.5x | 87.1/86.4 | 86.8/85.9 | 84.8/84.2 | 91.7 | 91.5 | 91.2 | 92.3 | 91.9 | 90.8 | 93.6 | 94.2 | 92.9 |
| 0.25x | 84.6/84.7 | 84.0/83.7 | 82.1/82.0 | 91.2 | 91.0 | 90.5 | 90.9 | 90.9 | 89.3 | 93.9 | 93.2 | 91.6 |

From Figure 3, the proposed DynaBERT and DynaRoBERTa achieve comparable accuracy as BERT$_{BASE}$ and RoBERTa$_{BASE}$, but often require fewer parameters, FLOPs or lower latency. Under the same efficiency constraint, sub-networks extracted from DynaBERT outperform DistilBERT and TinyBERT. Sub-networks extracted from DynaRoBERTa outperform LayerDrop by a large margin. They even consistently outperform LayerDrop trained with much more data. We speculate that it is because LayerDrop only allows flexibility in the depth direction. On the other hand, ours enables flexibility in both width and depth directions, which generates a significantly larger number of architectural configurations and better explores the balance between model accuracy and size.

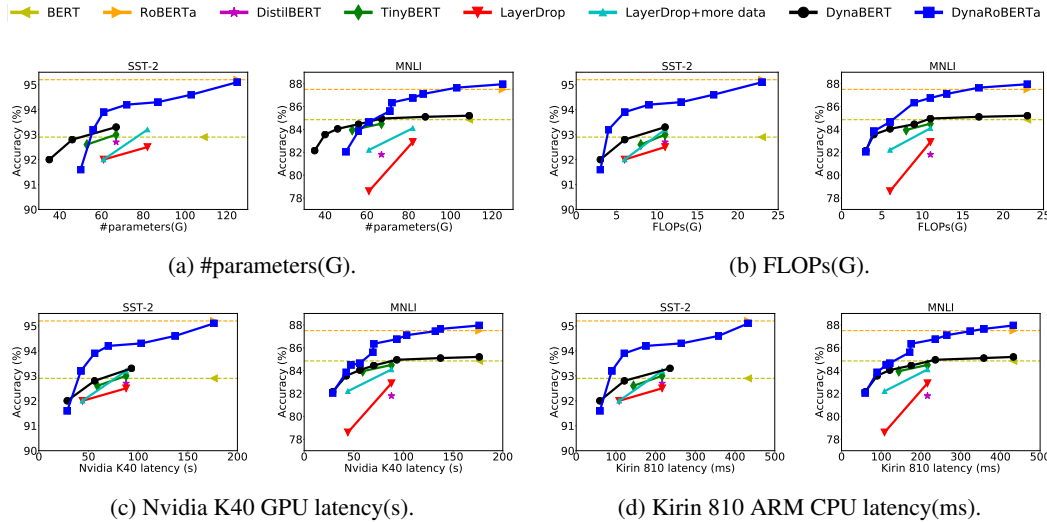

(a) #parameters(G).

(b) FLOPs(G).

(c) Nvidia K40 GPU latency(s).

(d) Kirin 810 ARM CPU latency(ms).

Figure 3: Comparison of #parameters, FLOPs, latency on GPU and CPU between our proposed DynaBERT and DynaRoBERTa and other methods. The GPU latency is the running time of 100 batches with batch size 128 and sequence length 128. The CPU latency is tested with batch size 1 and sequence length 128. Average accuracy of MNLI-m and MNLI-mm is plotted.

## 3.2 Results on SQuAD

SQuAD v1.1 (Stanford Question Answering Dataset) [19] contains 100k crowd-sourced question/answer pairs. Given a question and a passage, the task is to extract the start and end of the answer span from the passage. The performance metric used is EM (exact match) and F1.

Table 2 shows the results of sub-networks extracted from DynaBERT. Sub-network with only 1/2 width or depth of BERT$_{BASE}$ already achieves comparable or even better performance than it. Figure 4 shows the comparison of sub-networks of DynaBERT and other methods. We do not compare with LayerDrop [7] because SQuAD results are not reported in their paper. From Figure 4, with the same number of parameters, FLOPs, sub-networks extracted from DynaBERT outperform TinyBERT and DistilEBRT by a large margin.

Table 2: Development set results on SQuAD v1.1.

| Method | | SQuAD v1.1 | |
|---|---|---|---|
| BERT$_{BASE}$ | | 81.5/88.7 | |
| | $m_w$ \ $m_d$ | 1.0x | 0.75x | 0.5x |
| DynaBERT | 1.0x | 82.6/89.7 | 82.1/89.3 | 81.5/88.8 |
| | 0.75x | 82.3/89.5 | 82.1/89.3 | 80.9/88.5 |
| | 0.5x | 81.9/89.2 | 81.7/89.0 | 80.0/87.8 |
| | 0.25x | 80.7/88.1 | 79.9/87.5 | 76.6/85.0 |

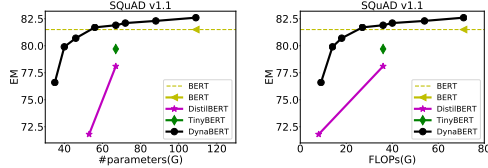

Figure 4: Comparison of #parameters and FLOPs between DynaBERT and other methods.

## 3.3 Ablation Study

**Training DynaBERT$_W$ with Adaptive Width.** In Table 3, we evaluate the importance of network rewiring, knowledge distillation and data augmentation (DA) in the training of DynaBERT$_W$ using the method in Section 2.1 on the GLUE benchmark. Due to space limit, only average accuracy of 4 width multipliers are shown in Table 3. Detailed accuracy for each width multiplier can be found in in Appendix C.2. DynaBERT$_W$ trained without network rewiring, knowledge distillation and data augmentation is called "vanilla DynaBERT$_W$". We also compare against the baseline of using separate networks, each of which is initialized from the BERT$_{BASE}$ with a certain width multiplier $m_w \in [1.0, 0.75, 0.5, 0.25]$, and then fine-tuned on the downstream task. From Table 3, vanilla DynaBERT$_W$ outperforms the separate network baseline. Interestingly, the performance gain is more obvious for smaller data sets CoLA, STS-B, MRPC and RTE. After network rewiring, the average accuracy increases by over 2 points. The average accuracy further increases by 1.5 points with knowledge distillation and data augmentation.

Table 3: Ablation study in training DynaBERT$_W$. Average accuracy of 4 width multipliers is reported.

| | MNLI-m | MNLI-mm | QQP | QNLI | SST-2 | CoLA | STS-B | MRPC | RTE | avg. |
|---|---|---|---|---|---|---|---|---|---|---|
| Separate network | 82.2 | 82.2 | 90.3 | 87.8 | 91.0 | 39.9 | 84.6 | 78.8 | 61.6 | 77.6 |
| Vanilla DynaBERT$_W$ | 82.2 | 82.5 | 90.6 | 89.1 | 91.2 | 44.0 | 87.4 | 80.5 | 64.2 | 79.0 |
| + Network rewiring | 83.1 | 83.0 | 90.9 | 90.4 | 91.7 | 51.4 | 89.1 | 83.8 | **69.7** | 81.4 |
| + Distillation and DA | **84.5** | **84.9** | **91.0** | **92.1** | **92.7** | **55.9** | **89.7** | **86.1** | 69.5 | 82.9 |

**Training DynaBERT with Adaptive Width and Depth.** In Table 4, we evaluate the effect of knowledge distillation, data augmentation and final fine-tuning in the training of DynaBERT described in Section 2.2. Detailed accuracy for each width and depth multiplier is in Appendix C.2. The DynaBERT trained without knowledge distillation, data augmentation and final fine-tuning is called "vanilla DynaBERT". From Table 4, with knowledge distillation and data augmentation, the average accuracy of each task is significantly improved compared to the vanilla counterpart on all three data sets. Additional fine-tuning further improves the performance on SST-2 and CoLA, but not MRPC. Empirically, we choose the model with higher average accuracy between before and after fine-tuning with the original data using the method described in Section 2.2.

**DynaBERT$_W$ as a "teacher assistant".** In Table 5, we also compare with directly distilling the knowledge from the rewired BERT to DynaBERT without DynaBERT$_W$. The average accuracy of 12 configurations of DynaBERT using DynaBERT$_W$ or not, are reported. As can be seen, using a width-adaptive DynaBERT$_W$ as a "teacher assistant" can efficiently bridge the large gap of size between the student and teacher, and has better performance on all three data sets investigated.

Table 4: Ablation study in training DynaBERT. Average accuracy of 12 configurations is reported.

|  | SST-2 | CoLA | MRPC |
|---|---|---|---|
| Vanilla DynaBERT | 91.3 | 46.0 | 82.1 |
| + Distillation and DA | 92.5 | 52.8 | **84.5** |
| + Fine-tuning | **92.7** | **54.8** | 83.2 |

Table 5: Whether using DynaBERT$_W$ as a "teacher assistant". Average accuracy of 12 configurations is reported.

|  | SST-2 | CoLA | MRPC |
|---|---|---|---|
| DynaBERT | 92.7 | 54.8 | 84.5 |
| - DynaBERT$_W$ | 92.3 | 54.1 | 84.4 |

**Adaptive Depth First or Adaptive Width First?** To finally obtain both width- and depth-adaptive DynaBERT, one can also train a only depth-adaptive model DynaBERT$_D$ first as the "teacher assistant", and then distill knowledge from it to DynaBERT. Table 6 shows the accuracy of DynaBERT$_W$ and DynaBERT$_D$ under different compression rates of the Transformer layers. As can be seen, DynaBERT$_W$ performs significantly better than DynaBERT$_D$ for smaller width/depth multiplier $0.5$. This may because unlike the width direction where the computation of attention heads and neurons are in parallel (Equations (1) and (2) in Section 2.1), the depth direction computes layer by layer consecutively. Thus we can not rewire the connections based on the importance of layers in DynaBERT$_D$, leading to severe accuracy drop of sub-networks with smaller depth in DynaBERT$_D$.

Table 6: Comparison of DynaBERT$_W$ and DynaBERT$_D$.

|  | QNLI | | | SST-2 | | | CoLA | | | STS-B | | | MRPC | | |
|---|---|---|---|---|---|---|---|---|---|---|---|---|---|---|---|
| $m_w$ or $m_d$ | 1.0x | 0.75x | 0.5x | 1.0x | 0.75x | 0.5x | 1.0x | 0.75x | 0.5x | 1.0x | 0.75x | 0.5x | 1.0x | 0.75x | 0.5x |
| DynaBERT$_W$ | 92.5 | 92.4 | 92.3 | 92.9 | 93.1 | 93.0 | 59.0 | 57.9 | 56.7 | 90.0 | 90.0 | 89.9 | 86.0 | 87.0 | 87.3 |
| DynaBERT$_D$ | 92.4 | 91.9 | 90.6 | 92.9 | 92.8 | 92.1 | 58.3 | 58.3 | 52.2 | 89.9 | 89.0 | 88.3 | 87.3 | 85.8 | 84.6 |

### 3.4 Looking into DynaBERT

We conduct a case study on the DynaBERT trained on `CoLA` by visualizing the attention distributions in Figure 5. The sentence used is "the cat sat on the mat." In [25, 11, 20], the attention heads for single-sentence task are found to mainly play "positional", "syntactic/semantic" functions. The positional head points to itself, adjacent tokens, [CLS], or [SEP] tokens, forming vertical or diagonal lines in the attention maps. The syntactic/semantic head points to tokens in a specific syntactic relation, and the attention maps do not have specific patterns.

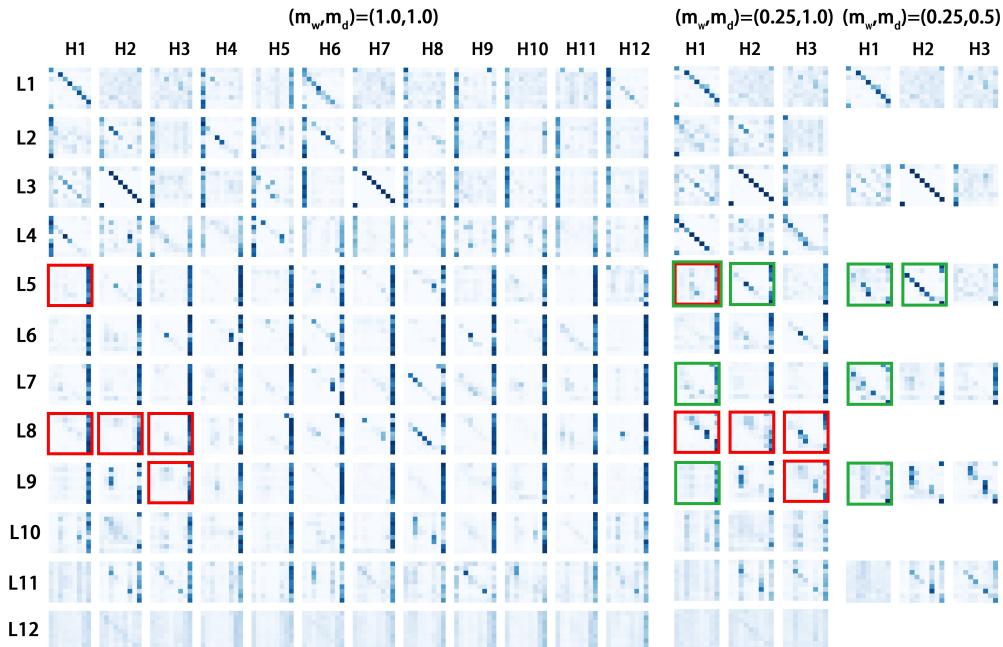

Figure 5: Attention maps of sub-networks with different widths and depths in DynaBERT trained on `CoLA`. The sentence used is "the cat sat on the mat."

From Figure 5, the attention patterns in the first three layers in the sub-network with $(m_w, m_d) = (0.25, 1.0)$ are quite similar to those in the full-sized model, while those at intermediate layers show a clear function fusion. For instance, H1 in L5, H1-3 in L8, H3 in L9 (marked with red squares) in the sub-network with $(m_w, m_d) = (0.25, 1.0)$ start to exhibit more syntactic or semantic patterns than their positional counterparts in the full-sized model. This observation is consistent with the finding in [9] that linguistic information is encoded in BERT's intermediate layers. Similarly, by comparing the attention maps in sub-networks with $(m_w, m_d) = (0.25, 1.0)$ and $(m_w, m_d) = (0.25, 0.5)$, functions (marked with green squares) also start to fuse when the depth is compressed.

Interestingly, we also find that DynaBERT improves the ability of distinguishing linguistic acceptable and non-acceptable sentences for CoLA. This is consistent with the superior performance of DynaBERT than BERT$_{\text{BASE}}$ in Table 1. The attention patterns of SST-2 also explain why it can be compressed by a large rate without severe accuracy drop. Details can be found in Appendix C.4.

### 3.5 Discussion

**Comparison of Conventional and Inplace Distillation.** To train width-adaptive CNNs, in [31], inplace distillation is used to boost the performance. Inplace distillation uses sub-network with the maximum width as the teacher while sub-networks with smaller widths in the same model as students. Training loss includes the loss from both the teacher network and the student network. Here we also adapt inplace distillation to train DynaBERT$_{\text{W}}$ in Table 7, and compare it with the conventional distillation used in Section 2.1. For inplace distillation, the student mimics the teacher and the teacher mimics a fixed fine-tuned task-specific BERT, via distillation loss over logits, embedding, and hidden states. From Table 7, inplace distillation has higher average accuracy on MRPC and RTE in training DynaBERT$_{\text{W}}$, but performs worse on three data sets after training DynaBERT.

Table 7: Comparison of conventional distillation and inplace distillation. Average accuracy of 4 width multipliers (DynaBERT$_{\text{W}}$) or 12 configurations (DynaBERT) is reported.

| Distillation type | | SST-2 | CoLA | MRPC | RTE | | SST-2 | CoLA | MRPC | RTE |
|---|---|---|---|---|---|---|---|---|---|---|
| Conventional | DynaBERT$_{\text{W}}$ | 92.7 | 55.9 | 86.1 | 69.5 | DynaBERT | 92.7 | 54.8 | 84.5 | 69.5 |
| Inplace [31] | | 92.6 | 55.9 | 87.0 | 70.0 | | 92.5 | 54.5 | 84.8 | 69.0 |

**Different Methods to Train DynaBERT$_{\text{W}}$.** For DynaBERT$_{\text{W}}$ in Section 2.1, we rewire the network only once, and train by alternating over four different width multipliers. In Table 8, we also adapt the following two methods in training width-adaptive CNNs to BERT: (1) using progressive rewiring (PR) as in [2] which progressively rewires the network as more width multipliers are supported; and (2) universally slimmable training (US) [31], which randomly samples some width multipliers in each iteration. The detailed setting of these two methods is in Appendix C.3. By comparing with Table 3, using PR or US has no significant difference from using the method in Section 2.1.

Table 8: Training DynaBERT$_{\text{W}}$ with PR and US. Average accuracy of 4 width multipliers is reported.

| | MNLI-m | MNLI-mm | QQP | QNLI | SST-2 | CoLA | STS-B | MRPC | RTE | avg. |
|---|---|---|---|---|---|---|---|---|---|---|
| PR [2] | 82.3 | 82.8 | 90.9 | 90.4 | 91.6 | 52.8 | 89.1 | 84.5 | 70.3 | 81.6 |
| US [31] | 82.6 | 82.9 | 90.6 | 90.3 | 91.5 | 51.2 | 89.1 | 83.8 | 69.6 | 81.3 |

## 4 Conclusion

In this paper, we propose DynaBERT which can flexibly adjust its size and latency by selecting sub-networks with different widths and depths. DynaBERT is trained by knowledge distillation. We adapt the width of the BERT model by varying the number of attention heads in MHA and neurons in the intermediate layer in FFN, and adapt the depth by varying the number of Transformer layers. Network rewiring is also used to make the more important attention heads and neurons shared by more sub-networks. Experiments on various tasks show that under the same efficiency constraint, sub-networks extracted from the proposed DynaBERT consistently achieve better performance than the other BERT compression methods.

## Broader Impact

Traditional machine learning computing relies on mobile perception and cloud computing. However, considering the speed, reliability, and cost of the data transmission process, cloud-based machine learning may cause delays in inference, user privacy leakage, and high data transmission costs. In such cases, in addition to end-cloud collaborative computing, it becomes increasingly important to run deep neural network models directly on edge. Recently, pre-trained language models like BERT have achieved impressive results in various natural language processing tasks. However, the BERT model contains tons of parameters, hindering its deployment to devices with limited resources. The difficulty of deploying BERT to these devices lies in two aspects. Firstly, the performances of various devices are different, and it is unclear how to deploy a BERT model suitable for each edge device based on its resource constraint. Secondly, the resource condition of the same device under different circumstances can be quite different. Once the BERT model is deployed to a specific device, dynamically selecting a part of the model for inference based on the device's current resource condition is also desirable.

Motivated by this, we propose DynaBERT. Instead of compressing the BERT model to a fixed size like existing BERT compression methods, the proposed DynaBERT can adjust its size and latency by selecting a sub-network with adaptive width and depth. By allowing both adaptive width and depth, the proposed DynaBERT also enables a large number of architectural configurations of the BERT model. Moreover, once the DynaBERT is trained, no further fine-tuning is required for each sub-network, and the benefits are threefold. Firstly, we only need to train one DynaBERT model, but can deploy different sub-networks to different hardware platforms based on their performances. Secondly, once one sub-network is deployed to a specific device, this device can select the same or smaller sub-networks for inference based on its dynamic efficiency constraints. Thirdly, different sub-networks sharing weights in one single model dramatically reduces the training and inference cost, compared to using different-sized models separately for different hardware platforms. This can reduce carbon emissions, and is thus more environmentally friendly.

Though not originally developed for compression, sub-networks of the proposed DynaBERT outperform other BERT compression methods under the same efficiency constraints like #parameters, FLOPs, GPU and CPU latency. Besides, the proposed DynaBERT at its largest size often achieves better performances as $BERT_{BASE}$ with the same size. A possible reason is that allowing adaptive width and depth increases the training difficulty and acts as regularization, and so contributes positively to the performance. In this way, the proposed training method of DynaBERT also acts as a regularization method that can boost the generalization performance.

Meanwhile, we also find that the compressed sub-networks of the learned DynaBERT have good interpretability. In order to maintain the representation power, the attention patterns of sub-networks with smaller width or depth of the trained DynaBERT exhibit function fusion, compared to the full-sized model. Interestingly, these attention patterns even explain the enhanced performance of DynaBERT on some tasks, e.g., enhanced ability of distinguishing linguistic acceptable and non-acceptable sentences for `CoLA`.

Besides the positive broader impacts above, since DynaBERT enables easier deployment of BERT, it also makes the negative impacts of BERT more severe. For instance, application in dialogue systems replaces help-desks and can cause job loss. Extending our method to generative models like GPT also faces the risk of generating offensive, biased or unethical outputs.

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
