[Supplementary Material]

# Appendix of DynaBERT: Dynamic BERT with Adaptive Width and Depth

## A   Layer Pruning Strategy and Hidden States Matching

In the training of DynaBERT with adaptive width and depth, when $m_d < 1$, we use the "Every Other" strategy in [2] and drop layers evenly to get a balanced network. Specifically, for depth multiplier $m_d$ (i.e., prune layers with a rate $1 - m_d$), we drop layers at depth $d$ which satisfies $\mathrm{mod}(d, \frac{1}{1-m_d}) \equiv 0$, because the lower layers in the student network which are found to change less from pre-training to fine-tuning [6]. We then match the hidden states of the remaining layers with those from all layers in the teacher model except those at depth $d$ which satisfies $\mathrm{mod}(d+1, \frac{1}{1-m_d}) \equiv 0$. In this way, we keep the knowledge learned in the last layer of the teacher network which is shown to be important in [12]. For a BERT model with 12 Transformer layers indexed by $1, 2, 3, \cdots, 12$, when $m_d = 0.75$, we drop the layers with indices $4, 8, 12$ of the student model. Then we match hidden states of the remaining 9 layers $L_S = \{1, 2, 3, 4, 5, 6, 7, 8, 9\}$ in the student with those indexed $L_T = \{1, 2, 4, 5, 6, 8, 9, 10, 12\}$ from the teacher network. When $m_d = 0.5$, we drop the layers indexed $2, 4, 6, 8, 10, 12$ of the student model. Then we match hidden states of the kept 6 layers $L_S = \{1, 2, 3, 4, 5, 6\}$ in the student with those indexed $L_T = \{2, 4, 6, 8, 10, 12\}$ from the teacher network. The loss $\ell_{hidn}$ is computed as

$$\ell'_{hidn}(\mathbf{H}^{(m_w, m_d)}, \mathbf{H}^{(m_w)}) = \sum_{l, l' \in L_S, L_T} \mathrm{MSE}(\mathbf{H}_l^{(m_w, m_d)}, \mathbf{H}_{l'}^{(m_w)}).$$

## B   More Experiment Settings

### B.1   Description of Data sets in the GLUE benchmark

The GLUE benchmark [11] is a collection of diverse natural language understanding tasks, including textual entailment (`RTE` and `MNLI`), question answering (`QNLI`), similarity and paraphrase (`MRPC`, `QQP`, `STS-B`), sentiment analysis (`SST-2`) and linguistic acceptability (`CoLA`). For `MNLI`, we use both the matched (`MNLI-m`) and mismatched (`MNLI-mm`) sections. We do not experiment on Winograd Schema (`WNLI`) because even a majority baseline outperforms many methods on it. We use the default train/validation/test splits from the official website[1].

### B.2   Hyperparameters

**GLUE benchmark.**   On the GLUE benchmark, the detailed hyperparameters for training DynaBERT$_\mathrm{W}$ in Section 2.1 and DynaBERT in Section 2.2 are shown in Table 1. The same hyperparameters as in Table 1 are used for DynaRoBERTa.

**SQuAD.**   Since $\mathcal{L}_{emb} + \mathcal{L}_{hidn}$ is several magnitudes larger than $\mathcal{L}_{pred}$ in this task, for both DynaBERT$_\mathrm{W}$ and DynaBERT, we separate the training into two stages, i.e., first using $\mathcal{L}_{emb} + \mathcal{L}_{hidn}$ as the objective and then $\mathcal{L}_{pred}$. When training with objective $\mathcal{L}_{emb} + \mathcal{L}_{hidn}$, we use the augmented data from [4] and train for 2 epochs. When training with objective $\mathcal{L}_{pred}$, we use the original training data and train for 10 epochs. The batch size is 12 throughout the training process. The other hyperparameters are the same as the GLUE benchmark in Table 1.

Table 1: Hyperparameters for different stages in training DynaBERT and DynaRoBERTa on the GLUE benchmark.

| | DynaBERT$_W$ | DynaBERT | |
|---|---|---|---|
| | Width-adaptive | Width- and depth-adaptive | Final fine-tuning |
| Width mutipliers | [1.0, 0.75,0.5,0.25] | [1.0, 0.75,0.5,0.25] | [1.0, 0.75,0.5,0.25] |
| Depth multipliers | 1 | [1.0, 0.75,0.5] | [1.0, 0.75,0.5] |
| Batch Size | 32 | 32 | 32 |
| Learning Rate | $2e-5$ | $2e-5$ | $2e-5$ |
| Warmup Steps | 0 | 0 | 0 |
| Learning Rate Decay | Linear | Linear | Linear |
| Weight Decay | 0 | 0 | 0 |
| Gradient Clipping | 1 | 1 | 1 |
| Dropout | 0.1 | 0.1 | 0.1 |
| Attention Dropout | 0.1 | 0.1 | 0.1 |
| Distillation | y | y | n |
| $\lambda_1, \lambda_2$ | 1.0, 0.1 | 1.0,1.0 | - |
| Data augmentation | y | y | n |
| Training Epochs (MNLI, QQP) | 1 | 1 | 3 |
| Training Epochs (other data sets) | 3 | 3 | 3 |

## B.3 FLOPs and Latency

To count the floating-point operations (FLOPs), we follow the setting in [1] and infer FLOPs with batch size 1 and sequence length 128. Unlike [1], we do not count the operations in the embedding lookup because the inference time in this part is negligible compared to that in the Transformer layers [10]. To evaluate the inference speed on GPU, we follow [4], and experiment on the QNLI training set with batch size 128 and sequence length 128. The numbers are the average running time of 100 batches on an Nvidia K40 GPU. To evaluate the inference speed on CPU, we experiment on Kirin 810 A76 ARM CPU with batch size 1 and sequence length 128.

## C   More Experiment Results

### C.1   More Results on the GLUE Benchmark

**Test Set Results.**    Table 2 shows the test set results. Again, the proposed DynaBERT achieves comparable accuracy as BERT$_{BASE}$ with the same size. Interestingly, the proposed DynaRoBERTa outperforms RoBERTa$_{BASE}$ on seven out of eight tasks. A possible reason is that allowing adaptive width and depth increases the training difficulty and acts as regularization, and so contributes positively to the performance.

Table 2: Test set results of the GLUE benchmark.

| | MNLI-m | MNLI-mm | QQP | QNLI | SST-2 | CoLA | STS-B | MRPC | RTE |
|---|---|---|---|---|---|---|---|---|---|
| BERT$_{BASE}$ | 84.6 | 83.6 | 71.9 | 90.7 | 93.4 | 51.5 | 85.2 | 87.5 | 69.6 |
| DynaBERT ($m_w, m_d = 1, 1$) | 84.5 | 84.1 | 72.1 | 91.3 | 93.0 | 54.9 | 84.4 | 87.9 | 69.9 |
| RoBERTa$_{BASE}$ | 86.0 | 85.4 | 70.9 | 92.5 | 94.6 | 50.5 | 88.1 | 90.0 | 73.0 |
| DynaRoBERTa ($m_w, m_d = 1, 1$) | 86.9 | 86.7 | 71.9 | 92.5 | 94.7 | 54.1 | 88.4 | 90.8 | 73.7 |

**Comparison with Other Methods on All GLUE Tasks.**    Figure 1 shows the comparison of our proposed DynaBERT and DynaRoBERTa with other compression methods on all GLUE tasks, under different efficiency constraints, including #parameters, FLOPs, latency on Nvidia K40 GPU and Kirin 810 A76 ARM CPU.

As can be seen, on all tasks, the proposed DynaBERT and DynaRoBERTa achieve comparable accuracy as BERT$_{BASE}$ and RoBERTa$_{BASE}$, but often require fewer parameters, FLOPs or lower latency. Similar to the observations in Section 3.1, under the same efficiency constraint, sub-networks extracted from our proposed DynaBERT outperform DistilBERT on all data sets except STS-B under #parameters, and outperforms TinyBERT on all data sets except MRPC; Sub-networks extracted from DynaRoBERTa outperform LayerDrop and even LayerDrop trained with much more data.

### C.2   Full Results of Ablation Study

**Training DynaBERT$_W$ with Adaptive Width.**    Table 3 shows the accuracy for each width multiplier in the ablation study in the training of DynaBERT$_W$. As can be seen, DynaBERT$_W$ performs similarly as the separate network baseline at its largest width and significantly better at smaller widths.

(a) #Parameters(G).

(b) FLOPs(G).

(c) Nvidia K40 GPU latency(s).

(d) Kirin 810 ARM CPU latency(ms).

Figure 1: Comparison of #parameters(G), FLOPs(G), Nvidia K40 GPU latency(s) and Kirin 810 ARM CPU latency(ms) between our proposed DynaBERT and DynaRoBERTa and other methods on the GLUE benchmark. Average accuracy of `MNLI-m` and `MNLI-mm` is plotted.

The smaller the width, the more significant the accuracy gain. From Table 3, after network rewiring, the average accuracy is over 2 points higher than the counterpart without rewiring. The accuracy gain is larger when the width of the model is smaller.

Table 3: Ablation study in the training of DynaBERT$_\text{W}$. Results on the development set are reported. The highest average accuracy of four width multipliers is highlighted.

| | $m_w$ | MNLI-m | MNLI-mm | QQP | QNLI | SST-2 | CoLA | STS-B | MRPC | RTE | avg. |
|---|---|---|---|---|---|---|---|---|---|---|---|
| | 1.0x | 84.8 | 84.9 | 90.9 | 92.0 | 92.9 | 58.1 | 89.8 | 87.7 | 71.1 | 83.6 |
| | 0.75x | 84.2 | 84.1 | 90.6 | 89.7 | 92.9 | 48.0 | 87.2 | 82.8 | 66.1 | 80.6 |
| Separate network | 0.5x | 81.7 | 81.7 | 89.7 | 86 | 91.4 | 37.2 | 84.5 | 75.5 | 55.2 | 75.9 |
| | 0.25x | 77.9 | 77.9 | 89.9 | 83.7 | 86.7 | 14.7 | 77.4 | 71.3 | 57.4 | 70.8 |
| | avg. | 82.2 | 82.2 | 90.3 | 87.8 | 91.0 | 39.9 | 84.6 | 78.8 | 61.6 | 77.6 |
| | 1.0x | 84.5 | 85.1 | 91.3 | 91.7 | 92.9 | 58.1 | 89.9 | 83.3 | 69.3 | 82.9 |
| | 0.75x | 83.5 | 84.0 | 91.1 | 90.1 | 91.7 | 54.5 | 88.7 | 82.6 | 65.7 | 81.3 |
| Vanilla DynaBERT$_\text{W}$ | 0.5x | 82.1 | 82.3 | 90.7 | 88.9 | 91.6 | 46.9 | 87.3 | 83.1 | 61 | 79.3 |
| | 0.25x | 78.6 | 78.4 | 89.1 | 85.6 | 88.5 | 16.4 | 83.5 | 72.8 | 60.6 | 72.6 |
| | avg. | 82.2 | 82.5 | 90.6 | 89.1 | 91.2 | 44.0 | 87.4 | 80.5 | 64.2 | 79.0 |
| | 1.0x | 84.9 | 84.9 | 91.4 | 91.6 | 91.9 | 56.3 | 90.0 | 84.6 | 70.0 | 82.8 |
| | 0.75x | 84.3 | 84.2 | 91.3 | 91.7 | 92.4 | 56.4 | 89.9 | 86.0 | 71.1 | 83.0 |
| + Network rewiring | 0.5x | 82.9 | 82.9 | 91.0 | 90.6 | 91.9 | 47.7 | 89.2 | 84.1 | 71.5 | 81.3 |
| | 0.25x | 80.4 | 80.0 | 90.0 | 87.8 | 90.4 | 45.1 | 87.3 | 80.4 | 66.0 | 78.6 |
| | avg. | 83.1 | 83.0 | 90.9 | 90.4 | 91.7 | 51.4 | 89.1 | 83.8 | **69.7** | 81.4 |
| | 1.0x | 85.1 | 85.4 | 91.1 | 92.5 | 92.9 | 59.0 | 90.0 | 86.0 | 70.0 | 83.5 |
| | 0.75x | 84.9 | 85.6 | 91.1 | 92.4 | 93.1 | 57.9 | 90.0 | 87.0 | 70.8 | 83.6 |
| + Distillation and DA | 0.5x | 84.4 | 84.9 | 91.0 | 92.3 | 93.0 | 56.7 | 89.9 | 87.3 | 71.5 | 83.4 |
| | 0.25x | 83.4 | 83.8 | 90.6 | 91.2 | 91.7 | 49.9 | 89.0 | 84.1 | 65.7 | 81.0 |
| | avg. | **84.5** | **84.9** | **91.0** | **92.1** | **92.7** | **55.9** | **89.7** | **86.1** | 69.5 | 82.9 |

**Training DynaBERT with Adaptive Width and Depth.**  Table 4 shows the accuracy for each width and depth multiplier in the ablation study in the training of DynaBERT.

Table 4: Ablation study in the training of DynaBERT. Results on the development set are reported. The highest average accuracy of four width multipliers for each depth multiplier is highlighted.

| | | SST-2 | | | CoLA | | | MRPC | | | RTE | | |
|---|---|---|---|---|---|---|---|---|---|---|---|---|---|
| | $m_w$\\$m_d$ | 1.0x | 0.75x | 0.5x | 1.0x | 0.75x | 0.5x | 1.0x | 0.75x | 0.5x | 1.0x | 0.75x | 0.5x |
| | 1.0x | 92.0 | 91.6 | 90.9 | 58.5 | 57.7 | 42.9 | 85.3 | 83.8 | 78.4 | 67.9 | 66.8 | 66.4 |
| Vanilla DynaBERT | 0.75x | 92.3 | 91.6 | 91.1 | 57.9 | 56.4 | 42.4 | 86.0 | 83.1 | 78.7 | 69.0 | 66.8 | 63.9 |
| | 0.5x | 91.9 | 91.9 | 90.6 | 55.9 | 53.3 | 40.6 | 86.0 | 83.1 | 79.7 | 68.2 | 65.0 | 63.9 |
| | 0.25x | 91.6 | 91.3 | 89.0 | 52.0 | 50.0 | 27.6 | 83.1 | 80.4 | 77.5 | 65.3 | 63.5 | 60.3 |
| | avg. | 92.0 | 91.6 | 90.4 | 56.1 | 54.4 | 38.4 | 85.1 | 82.6 | 78.6 | 67.6 | 65.5 | 63.6 |
| | $m_w$\\$m_d$ | 1.0x | 0.75x | 0.5x | 1.0x | 0.75x | 0.5x | 1.0x | 0.75x | 0.5x | 1.0x | 0.75x | 0.5x |
| | 1.0x | 92.9 | 93.3 | 92.7 | 57.1 | 56.7 | 52.6 | 86.3 | 85.8 | 85.0 | 72.2 | 70.4 | 66.1 |
| + Distillation and | 0.75x | 93.1 | 93.1 | 92.1 | 57.7 | 55.4 | 51.9 | 86.5 | 85.5 | 84.1 | 72.6 | 72.2 | 64.6 |
| Data augmentation | 0.5x | 92.9 | 92.1 | 91.3 | 54.1 | 53.7 | 47.5 | 84.8 | 84.1 | 83.1 | 72.9 | 72.6 | 66.1 |
| | 0.25x | 92.5 | 91.7 | 91.6 | 50.7 | 51.0 | 44.6 | 83.8 | 83.8 | 81.4 | 67.5 | 67.9 | 62.5 |
| | avg. | 92.9 | 92.6 | 91.9 | 54.9 | 54.2 | 49.2 | **85.4** | **84.8** | **83.4** | **71.3** | 70.8 | 64.8 |
| | $m_w$\\$m_d$ | 1.0x | 0.75x | 0.5x | 1.0x | 0.75x | 0.5x | 1.0x | 0.75x | 0.5x | 1.0x | 0.75x | 0.5x |
| | 1.0x | 93.2 | 93.3 | 92.7 | 59.7 | 59.1 | 54.6 | 84.1 | 83.6 | 82.6 | 72.2 | 71.8 | 66.1 |
| + Fine-tuning | 0.75x | 93.0 | 93.1 | 92.8 | 60.8 | 59.6 | 53.2 | 84.8 | 83.6 | 82.8 | 71.8 | 73.3 | 65.7 |
| | 0.5x | 93.3 | 92.7 | 91.6 | 58.4 | 56.8 | 48.5 | 83.6 | 83.3 | 82.6 | 72.2 | 72.2 | 67.9 |
| | 0.25x | 92.8 | 92.0 | 92.0 | 50.9 | 51.6 | 43.7 | 82.6 | 83.6 | 81.1 | 68.6 | 68.6 | 63.2 |
| | avg. | **93.1** | **92.8** | **92.3** | **57.5** | **56.8** | **50.0** | 83.8 | 83.5 | 82.3 | 71.2 | **71.5** | **65.7** |

## C.3  Full Results of Different Methods to Train DynaBERT$_\text{W}$

**Progressive Rewiring.**  Instead of rewiring the network only once before training, "progressive rewiring" progressively rewires the network as more width multipliers are supported throughout the training. Specifically, for four width multipliers $[1.0, 0.75, 0.5, 0.25]$, progressive rewiring first sorts the attention heads and neurons and rewires the corresponding connections before training to support width multipliers $[1.0, 0.75]$. Then the attention heads and neurons are sorted and the network is rewired again before supporting $[1.0, 0.75, 0.5]$. Finally, the network is again sorted and rewired before supporting all four width multipliers. For "progressive rewiring", we tune the initial learning rate from $\{2e-5, 1e-5, 2e-5, 5e-6, 2e-6\}$ and pick the best-performing initial learning rate $1e-5$. Table 5 shows the development set accuracy on the GLUE benchmark for using progressive rewiring. Since progressive rewiring requires progressive training and is time-consuming, we do not use data augmentation and distillation. We use cross-entropy loss between predicted labels and the

ground-truth labels as the training loss. By comparing with Table 3 in Section 3.3, using progressive rewiring has no significant gain over rewiring only once.

Table 5: Training DynaBERT$_W$ using progressive rewiring (PR).

| $m_w$ | MNLI-m | MNLI-mm | QQP | QNLI | SST-2 | CoLA | STS-B | MRPC | RTE | avg. |
|---|---|---|---|---|---|---|---|---|---|---|
| 1.0x | 84.6 | 84.5 | 91.5 | 91.6 | 92.4 | 57.4 | 90.1 | 86.5 | 70.0 | 83.2 |
| 0.75x | 83.6 | 84.0 | 91.2 | 91.4 | 91.7 | 56.6 | 89.7 | 84.8 | 70.8 | 82.6 |
| 0.5x | 82.5 | 82.9 | 91.0 | 90.8 | 91.9 | 52.2 | 89.1 | 84.1 | 72.9 | 81.9 |
| 0.25x | 78.3 | 79.7 | 89.9 | 87.9 | 90.4 | 45.1 | 87.6 | 82.4 | 67.5 | 78.8 |
| avg. | 82.3 | 82.8 | 90.9 | 90.4 | 91.6 | 52.8 | 89.1 | 84.5 | 70.3 | 81.6 |

**Universally Slimmable Training.** Instead of using a pre-defined list of width multipliers, universally slimmable training [13] samples several width multipliers in each training iteration. Following [13], we also use inplace distillation for universally slimmable training. For universally slimmable training, we tune $(\lambda_1, \lambda_2)$ in $\{(1, 1), (1, 0), (0, 1), (1, 0.1), (0.1, 1), (0.1, 0.1)\}$ on MRPC and choose the best-performing one $(\lambda_1, \lambda_2) = (0.1, 0.1)$. The corresponding results for can be found in Table 6. For better comparison with using pre-defined width multipliers, we also report results when the width multipliers are $[1.0, 0.75, 0.5, 0.25]$. We do not use data augmentation here. By comparing with Table 3 in Section 3.3, there is no significant difference between using universally slimmable training and the alternative training as used in Algorithm 1.

Table 6: Training DynaBERT$_W$ using universally slimmable training (US).

| $m_w$ | MNLI-m | MNLI-mm | QQP | QNLI | SST-2 | CoLA | STS-B | MRPC | RTE | avg. |
|---|---|---|---|---|---|---|---|---|---|---|
| 1.0x | 84.6 | 85.0 | 91.2 | 91.7 | 92.4 | 59.7 | 90.0 | 85.3 | 69.0 | 83.2 |
| 0.75x | 84.0 | 84.5 | 91.1 | 91.3 | 92.5 | 56.7 | 90.0 | 85.3 | 70.4 | 82.9 |
| 0.5x | 82.2 | 82.6 | 90.7 | 90.5 | 91.1 | 52.1 | 89.2 | 85.3 | 71.5 | 81.7 |
| 0.25x | 79.7 | 79.5 | 89.3 | 87.5 | 90.1 | 36.4 | 87.3 | 79.4 | 67.5 | 77.4 |
| avg. | 82.6 | 82.9 | 90.6 | 90.3 | 91.5 | 51.2 | 89.1 | 83.8 | 69.6 | 81.3 |

## C.4 Looking into DynaBERT

**CoLA.** CoLA is abbreviated for the "Corpus of Linguistic Acceptability" and is a binary single-sentence classification task, where the goal is to predict whether an English sentence is linguistically "acceptable". Figure 2 shows the attention maps of the learned DynaBERT with two different width multipliers $m_w = 1.0$ and $0.25$. We use both a linguistically acceptable sentence "the cat sat on the mat." and a non-acceptable one ".mat the on sat cat the" whose words are in the reverse order. As can be seen, in the last two Transformer layers of DynaBERT of both widths, for the linguistic non-acceptable sentence, the attention heads do not encode useful information, with each word attending to every other word with almost equal probability. Figure 3 shows the attention maps obtained by BERT$_{BASE}$ fine-tuned on CoLA, with the same linguistic acceptable and non-acceptable sentence as in Figure 2. As can be seen, unlike DynaBERT, the attention maps in the final two layers still show positional or syntactic patterns. This observation reveals the enhanced ability of the proposed DynaBERT in distinguishing linguistic acceptable and non-acceptable sentences. Similar observations are also found in other samples in CoLA data set.

**SST-2.** SST-2 (the Stanford Sentiment Treebank) is a binary single-sentence classification task consisting of sentences extracted from movie reviews with human annotations of their sentiment. Figure 4 shows the attention maps obtained by DynaBERT with annotations of both positive and negative sentiment. The sentence with positive sentiment is "a smile on your face.". The sentence with negative sentiment is "an extremely unpleasant film .". As can be seen, for both $m_w = 1$ and $0.25$, most attention maps in the final few layers point to the last token "[SEP]", which is not used in the downstream task. This indicates that there is redundancy in the Transformer layers. This is also consistent with the finding in Section 3.1 that, even when the depth multiplier is only $m_d = 0.5$ (i.e., 6 Transformer layers), the model has only less than 1 point of accuracy degradation for both widths.

(a) "the cat sat on the mat."

(b) ".mat the on sat cat the"

Figure 2: Attention maps in sub-networks with different widths in DynaBERT trained on CoLA.

(a) "the cat sat on the mat."

(b) ".mat the on sat cat the"

Figure 3: Attention maps in BERT$_{\text{BASE}}$ fine-tuned on CoLA.

(a) "a smile on your face."

(b) "an extremely unpleasant film."

Figure 4: Attention maps in sub-networks with different widths in DynaBERT trained on `SST-2`.

## D   Related Work on the Capacity of Language Models

There are also related works that study the relationship between the capacity and performance of language models. It is shown in [6, 8] that considerable redundancy and over-parametrization exists in BERT models. In [3], it is shown that the BERT's layers encode a hierarchy of linguistic information, with surface features at the bottom, syntactic features in the middle and semantic features at the top. The capacity of other language models besides BERT like character CNN and recurrent networks are also studied in [5, 7]. In [9], it is shown that pre-trained language models with moderate sized representations are able to recover arbitrary sentences.

## E   Preliminary Results of Applying DynaBERT in the Pretraining Phase

In this section, we use the proposed method for pre-training a BERT with adaptive width and depth. We use a pre-trained 6-layer BERT downloaded from the official Google BERT repository `https://github.com/google-research/bert` as the backbone model. To make sub-networks of DynaBERT the same size as those small models, for width, we also adapt the hidden state size $H = 128, 256, 512, 768$ besides attention heads and intermediate layer neurons. For depth, we adjust the number of layers to be $L = 4, 6$. Distillation loss over the hidden states in the last layer is used as the training objective. The number of training epochs is 5. After pre-training DynaBERT, we fine-tune each separate sub-network with the original task-specific data on MNLI-m and report the development set results in Table 7. We compare with separately pre-trained small models in Google BERT repository. As can be seen, sub-networks of the pre-trained DynaBERT outperform separately pre-trained small networks.

Table 7: Development set accuracy on MNLI-m of separately pre-trained BERT models and sub-networks of a pre-trained DynaBERT.

| (L, H) | (6, 768) | (6, 512) | (6, 256) | (6, 128) | (4, 768) | (4, 512) | (4, 256) | (4, 128) |
|---|---|---|---|---|---|---|---|---|
| Separate small networks | 81.8 | 80.3 | 76.0 | 72.4 | 80.1 | 78.6 | 74.9 | 70.7 |
| Sub-networks of DynaBERT | 82.0 | 81.0 | 77.8 | 73.0 | 81.5 | 80.4 | 76.1 | 71.4 |

## Footnotes

[1]`https://gluebenchmark.com/tasks`