[Reviews · NeurIPS 2020]

Review 1

Summary and Contributions: This paper presents DynaBERT which adapts the size of a BERT or RoBERTa model both in width and in depth. While the depth adaptation is well known, the width adaptation uses importance scores for the heads to rewire the network, so the most useful heads are kept. The authors show that this approach, combined with a procedure to distill knowledge from a vanilla large pre-trained and fine-tuned model into the smaller adapted one can perform very well when compared to both the original large model, and to other methods used to compress BERT-like models. This work introduces a novel rewiring approach to reduce the width of the transformer model by a particular multiplier and combines several other previously known approaches (adaptive depth, distillation) to achieve superior results.

Strengths: The main strength of the work is in the empirical evaluation of the results and the various ablation studies to delineate contributions of the different techniques proposed in the paper that together make up DynaBERT. The width-shrinking technique appears to be novel and works very well, which will hopefully inspire additional research. I believe this work will be useful to the NeurIPS community.

Weaknesses: One weakness is that the procedure to obtain a compressed model is quite involved, and must be done separately for each downstream task. It would be interesting to see whether this approach can be adapted to work during the pre-training phase.

Correctness: The claims and methodology seem correct from what is described in the paper.

Clarity: The paper is very well written and is quite easy to read!

Relation to Prior Work: This work builds on many techniques introduced in other works. I believe it did a good job delineating its contribution.

Reproducibility: Yes

Additional Feedback: I've read the author rebuttal and thank the authors for their clarifications. I am believe my rating is still appropriate for this work.


Review 2

Summary and Contributions: Various works have proposed decreasing the depth of BERT style transformer models. This work proposes a way to dynamically decreases both depth and width by first training a model with dynamic width, then distilling from that model to a model with both dynamic width and depth. In terms of parameter efficiency and inference speed this approach outperforms other ways of distilling BERT.

Strengths: The empirical results are strong, and present an interesting analysis of whether depth of width is the most important. The combination of depth and width reduction is novel, along with the two stage procedure. Interesting analysis of important attention heads.

Weaknesses: I found the paper quite dense and hard to read. The results are impressive, but rely on various complicated procedures (data augmentation, multiple rounds of finetuning and distillation.)

Correctness: Yes.

Clarity: I found the paper quite dense, and would appreciate more precise definitions of various tricks. E.g. Table 4, what exactly is 'fine-tuning'? You should link more between your results section and your methods section, i.e. say something like 'this result is from the method described in section 3.1' or whatever the case may be. line 140: 'In this work, we use the model that has higher 141 average validation accuracy of all sub-networks, between with and without fine-tuning.' I can't work out what this sentence means.

Relation to Prior Work: Yes.

Reproducibility: Yes

Additional Feedback: I believe equation 1 is normally expressed in terms of concatenating together attention heads, which confused me at first. You might want to be clearer about why you are using a different formulation of the standard transformer layer for your purposes. I think you need to briefly define TinyBERT and LayerDrop in case the reader isn't familiar. Why are they fair comparison points (i.e. maybe they require less fine-tuning compute power, given your multi-step process)? I would also briefly define your data augmentation procedure, even if it is described in prior work. There are a lot of moving parts here, and you perform ablation studies to show they all help. But something to think about is if there is a simpler method that can give you similar results. You should think about potential negative consequences in your broader impact study, for example the complexity of the distillation procedure might make it harder to apply to new domains (if we need to tune many hyperparameters etc.). Update: I thank the authors for their useful response to my questions. I will increase my score given most of my concerns were adressed, but I urge the authors to pay attention to readability and clarity for the final version.


Review 3

Summary and Contributions: Authors propose DynaBERT which allows a user to adjusts size and latency based on adaptive width and depth of the BERT model. They show that they can do this, offering the user choices based on what the user would need. They show results that seem to indicate performance gains at similar sizes (in terms of width and depth) in comparison to existing compression methods for BERT.

Strengths: The experimentation is comprehensive and addresses a problem in general distillation where the user doesn't have a method that gives them complete control. The paper is written well and is mostly clear.

Weaknesses: The broader impacts and related work sections are a bit weak. The broader impacts reads as a rehashing of the conclusion and the related work part is integrated into the introduction and given just a half paragraph. I'd like to see these improved. See the additional feedback section.

Correctness: I think all of these aspects are correct and clean.

Clarity: Paper is well written.

Relation to Prior Work: Kind of. Its clear how it relates to some of the other fixed distillation approaches, but without a comprehensive related work section its hard to figure out.

Reproducibility: Yes

Additional Feedback: Random things: - Table 1 is a bit overloaded and difficult to parse. Also I'm not sure which row and column are m_w vs m_d. - Figure 3 is really difficult to parse too, RoBERTa performance is covered by the DynaRoBERTa and sometimes similarly for BERT. Can you present this differently with lines corresponding to the base models? - Why MNLI and SST-2 specifically for Figure 3 rather than others? Related Work: There's a little bit of discussion in the first half of paragraph 2 of the introduction, but no comprehensive addressing of how your work sits in context to the work already out there. Including work that talks about the capacity of large language models, what they can and can't do would be important here, how more layers/parameters help language models in general (Jawahar et al 2019; What does BERT learn about the structure of language?, Jozefowicz et al 2016 Exploring the limits of language modeling, Melis et al 2017 On the State of the Art of Evaluation in Neural Language Models, Subramani et al 2019 Can Unconditional Language Models Recover Arbitrary Sentences?). There are many others but this is a subset. Experimentation & Analysis: The discussion section and analysis section list out results and present tables, but there isnt much of a discussion in the main body. Can you talk about and put your results in context so that its easier for the reader to understand how and why certain approaches work? I know its difficult and sometimes cost prohibitive, but having statistical significance on your studies would go a long way into seeing if there really is a statistical difference between your approach and others. Broader Impacts: This reads like a rehashing, extra hashing of your conclusion rather than really discussing ethical and societal implications of your work. I'd like to see quite a bit here discussing the potentials of this method in helping and harming if used in certain ways. ------------------------------------------------------------------------------------------ Thanks for the authors for submitting an author response. I read through the others' reviews and the author response. Thank you for addressing some of the related work and statistical significance concerns. The clarity around how this relates and is put into context around other prior work would help me make better judgment. The statement you made connecting this to Jawahar work and the stability of the system is a positive. I was between a 4 and a 5 before the author response and am now I'm at around a 5.5, leaning slightly away from acceptance, so I'm gonna stick to the 5.

[Author Response · NeurIPS 2020]

Genral response: We thank all reviewers for their constructive comments. Below is our response for common questions.

**Q1. link more between the results section and your methods (R2 & R3):** Thanks for the suggestion. We will

reorganize the method and experiment sections, and add more links between them in the next version.

**Q2. broader impact (R2 & R3):** For the **positive** side, as is detailed in the Broader Impact section, DynaBERT (i)

alleviates concerns about the privacy by moving computation to edge; (ii) enables flexible deployment scenarios of

BERT models; and (iii) is more environmentally friendly due to weight sharing. For the **negative** side, DynaBERT

enables easier deployment of BERT, and thus makes the negative impacts of BERT more severe, e.g., application in

dialogue systems replaces help-desks and can cause job loss. Extending our method to generative models like GPT also

faces risk of generating offensive, biased or unethical outputs. We will detail these impacts in the next version.

Reviewer 1 **Q1.“whether this approach can be adapted to work during the pre-training phase”**: Below we show

results of using the proposed method for pre-training. Due to time limit, we only vary the width and depth of a 6-layer

BERT. We compare with separately pre-trained small models in Google BERT repository (`https://github.com/`

`google-research/bert`) and report the accuracy after fine-tuning on MNLI-m. To make sub-networks of DynaBERT

the same size as those small models, for width, we also adapt the hidden state size $H = 128, 256, 512, 768$ besides

attention heads and intermediate layer neurons. For depth, we adjust the number of layers to be $L = 4, 6$. As can be

seen, the sub-networks of the pre-trained DynaBERT outperform separately pre-trained small networks.

| (L, H) | (6, 768) | (6, 512) | (6, 256) | (6, 128) | (4, 768) | (4, 512) | (4, 256) | (4, 128) |
|---|---|---|---|---|---|---|---|---|
| Dev set accuracy of separate small networks | 81.8 | 80.3 | 76.0 | 72.4 | 80.1 | 78.6 | 74.9 | 70.7 |
| Dev set accuracy of sub-networks of DynaBERT | 82.0 | 81.0 | 77.8 | 73.0 | 81.5 | 80.4 | 76.1 | 71.4 |

Reviewer 2 **Q1.“paper quite dense and hard to read...rely on various complicated procedures”, “if there is a**

**simpler method”**: We also tried to simplify the current approach, but ablation study in Section 3.3 shows that all these

various procedures helps performance and can not be removed. We will continue thinking about simplifying the method.

**Q2.“Table 4, what exactly is ’fine-tuning’?”** :This is the ‘fine-tuning’ mentioned in Lines 138-139 in Section 2.2.

**Q3.“line 140: ‘In this work... I can’t work out what this sentence means.”**: The training in Section 2.2 consists of

two parts: (i) training with augmented data and distillation objective (Lines 123-138); and (ii) fine tuning with original

data and cross-entropy loss (Lines 138-141). Here ‘without finetuning’ means using only (i), while ‘with finetuning’

means using first (i) and then (ii). In the experiments in Table 1 in Section 3.1, we report the best results among all the

sub-networks produced by either (i) or (ii).

**Q4. different formulation of the standard transformer layer in equation (1)**: Equation (1) shows that the attention

heads can be computed in parallel and thus can be used to adjust the width of a Transformer layer.

**Q5.“define TinyBERT and LayerDrop ...Why are they fair comparison points (i.e. maybe they require less**

**fine-tuning compute power...)? ”** : We will add descriptions for TinyBERT and LayerDrop. They are popular BERT

compression methods, and are computationally more expensive than ours as they need to redo the pretraining step. We

compare with them to show that sub-networks of DynaBERT outperform similar-sized models.

**Q6.“briefly define your data augmentation procedure”**: We will add it in the final version.

**Q7.“the complexity of the distillation procedure might make it harder to apply to new domains (if we need to**

**tune many hyperparameters etc.)”**: The hyperparameters are easy and cheap to be determined. We use only a few

samples to estimate the magnitude of different distillation losses, then choose $\lambda_1, \lambda_2$ to make them have similar scale.

Reviewer 3 **Q1.“Table 1 is a bit overloaded and difficult to parse...which row and column are** $m_w$ **vs** $m_d$**”**: The

row is depth multiplier $m_d$ taking 3 values while column is width multiplier $m_w$ taking 4 values, as defined in Line 150.

**Q2.“Figure 3 is really difficult to parse too... Can you present this differently with lines corresponding to the**

**base models?”** : We will replace the markers with lines for RoBERTa and BERT base as suggested in the final version.

**Q3.“Why MNLI and SST-2 specifically for Figure 3 rather than others?”** Due to space limit, we only show plots

for MNLI and SST-2 in the main content, and put those for the others in Appendix C.1 as mentioned in Line 177.

**Q4. related work**: (**1**) The primary goal of our paper is training multiple compressed sub-networks in the same model

by varying width and depth. Thus we first discuss related work on compressing Transformer-based models to a certain

size or various sizes by adapting only depth in Paragraph 2. Then we discuss the connection/difference between our

method and others in Paragraph 3. (**2**) Thanks for providing references about the capacity of language models. In our

paper, we also empirically studied capacity of DynaBERT in Section 3. From Table 1, CoLA (the task of linguistic

acceptability judgments) is relatively more sensitive to the capacity. Figure 5 also shows that when reducing capacity for

CoLA, the function fusion of attention heads occurs mainly in the intermediate layers. This is consistent with the finding

in the recommended reference (Jawahar et al 2019) that, BERT’s intermediate layers encode linguistic information. The

other 3 references studied different models (e.g., character CNN and recurrent networks in Jozefowicz et al 2016; Melis

et al 2017) or tasks (i.e. generation in Subramani et al 2019), and will also be discussed in the final version.

**Q5. statistical significance**: Statistical significance needs many runs of both our method and others. This is infeasible

due to limited time of rebuttal. Below we report $mean \pm std$ accuracy from 5 repetitions on STS-B and SST-2, and

leave more rigorous comparison for more tasks as a future work. The small std indicates the stability of DynaBERT.

| $(m_w, m_d)$ | (1x, 1x) | (1x, 0.75x) | (1x, 0.5x) | (0.75x, 1x) | (0.75x, 0.75x) | (0.75x, 0.5x) | (0.5x, 1x) | (0.5x, 0.75x) | (0.5x, 0.5x) | (0.25x, 1x) | (0.25x, 0.75x) | (0.25x, 0.5x) |
|---|---|---|---|---|---|---|---|---|---|---|---|---|
| STS-B | $89.96 \pm 0.08$ | $89.39 \pm 0.08$ | $88.51 \pm 0.11$ | $89.86 \pm 0.08$ | $89.30 \pm 0.08$ | $88.46 \pm 0.09$ | $89.75 \pm 0.04$ | $89.19 \pm 0.07$ | $88.28 \pm 0.09$ | $89.16 \pm 0.06$ | $88.32 \pm 0.10$ | $87.04 \pm 0.04$ |
| SST-2 | $93.11 \pm 0.34$ | $93.16 \pm 0.16$ | $92.65 \pm 0.17$ | $93.14 \pm 0.42$ | $92.93 \pm 0.35$ | $92.51 \pm 0.26$ | $92.95 \pm 0.23$ | $92.81 \pm 0.20$ | $91.65 \pm 0.13$ | $92.64 \pm 0.34$ | $92.30 \pm 0.27$ | $91.89 \pm 0.38$ |

[Meta-Review · NeurIPS 2020]

Two reviewers gave strong accept ratings, and the third gave a borderline rating (in the discussion). The reviewers noted the strong empirical results, as well as the novelty of the proposed approach, building on existing works which are well discussed. One minor concern is the complexity of the approach (but as noted by one reviewers, existing methods are also somewhat complicated). I agree with the reviewers, and therefore, the paper is accepted.